# A Real-World, Observational, Prospective Study to Assess the Molecular Epidemiology of Epidermal Growth Factor Receptor (*EGFR*) Mutations upon Progression on or after First-Line Therapy with a First- or Second-Generation EGFR Tyrosine Kinase Inhibitor in *EGFR* Mutation-Positive Locally Advanced or Metastatic Non-Small Cell Lung Cancer: The ‘LUNGFUL’ Study

**DOI:** 10.3390/cancers13133172

**Published:** 2021-06-25

**Authors:** Giannis Mountzios, Anna Koumarianou, Alexandros Bokas, Dimitrios Mavroudis, Epaminondas Samantas, Evangelos Georgios Fergadis, Helena Linardou, Panagiotis Katsaounis, Elias Athanasiadis, Michalis V. Karamouzis, George Pentheroudakis, Sofia Lampaki, Marios E. Froudarakis, Eleni-Isidora A. Perdikouri, Alvertos Somarakis, Foteini Papageorgiou, Zoe Paparepa, Aristeidis Nikolaou, Konstantinos N. Syrigos

**Affiliations:** 1Fourth Oncology Department and Clinical Trials Unit, Henry Dunant Hospital Center, 11526 Athens, Greece; 2Hematology-Oncology Unit, Fourth Department of Internal Medicine, Attikon University Hospital, National and Kapodistrian University of Athens, 12462 Athens, Greece; akoumari@yahoo.com; 3First Department of Clinical Oncology, Theagenio Cancer Hospital, 54007 Thessaloniki, Greece; a.pathologiki@theagenio.gov.gr; 4Department of Medical Oncology, University Hospital of Heraklion, 71110 Crete, Greece; mavrudis@med.uoc.gr; 52nd Oncology Department, Metropolitan Hospital, 18547 Athens, Greece; epsam@otenet.gr; 6Medical Oncology Department, Metaxa Cancer Hospital, 18537 Piraeus, Greece; e.fergadis@windowslive.com; 74th Oncology Department & Comprehensive Clinical Trials Center, Metropolitan Hospital, 18547 Athens, Greece; elinardou@metropolitan-hospital.gr; 8Oncology Department, Metropolitan Hospital, 15562 Athens, Greece; pvkatsaounis@gmail.com; 9Department of Medical Oncology, Mitera Hospital, 15123 Athens, Greece; iathanasiadis@hygeia.gr; 10Molecular Oncology Unit, Department of Biological Chemistry Medical School, 11525 Athens, Greece; m.karamouzis@imibe.org; 11Department of Medical Oncology, Medical School University of Ioannina, 45500 Ioannina, Greece; gpenther@otenet.gr; 12Pulmonary Department, General Hospital ‘G. Papanikolaou’, Aristotle University of Thessaloniki, 57010 Thessaloniki, Greece; pneumon-kliniki@auth.gr; 13Department of Respiratory Medicine, Medical School of Alexandroupolis Democritus University of Thrace, 68100 Alexandroupolis, Greece; mfroud@med.duth.gr; 14Oncology Department, General Hospital ‘Papageorgiou’, Aristotle University of Thessaloniki, 56429 Thessaloniki, Greece; cpapandreou@auth.gr; 15Medical Affairs Department, AstraZeneca, 15123 Athens, Greece; alvertos.somarakis@astrazeneca.com (A.S.); aristeidis.nikolaou@astrazeneca.com (A.N.); 16Diagnostics Function Oncology, AstraZeneca, 15123 Athens, Greece; Fotini.Papageorgiou@astrazeneca.com; 17Clinical Operations, AstraZeneca, 15123 Athens, Greece; Zoi.Paparepa@astrazeneca.com; 18Third Department of Medicine, National and Kapodistrian University of Athens, School of Medicine, Sotiria Hospital, 11527 Athens, Greece; ksyrigos@med.uoa.gr

**Keywords:** biopsy, carcinoma, non-small-cell lung cancer, EGFR-tyrosine kinase inhibitor, epidermal growth factor receptor, T790M mutation

## Abstract

**Simple Summary:**

Non-small cell lung cancer (NSCLC) accounts for approximately 85% of lung cancer cases, with few patients carrying driver mutations in the gene encoding for epidermal growth factor receptor (*EGFR*). Advances in translational research have established EGFR tyrosine kinase inhibitors (TKIs) as the standard first-line therapy for NSCLC patients with activating *EGFR* mutations. The aim of our observational study was to assess the frequency of T790M acquired resistance and predictors of its presence, in patients with *EGFR*-mutated locally advanced or metastatic NSCLC who have progressed in the first-line EGFR-TKI treatment setting with first- or second-generation TKIs and have undergone molecular testing in tissue and/or plasma biopsy. The study highlights the challenges of performing tissue re-biopsy in routine care settings, which can lead to patients considered non-eligible for certain therapies from which they can benefit, and merits further actions from the healthcare community, in order to establish re-biopsy as a standard procedure.

**Abstract:**

*Background*: Real-world data on the molecular epidemiology of *EGFR* resistance mutations at or after progression with first- or second-generation EGFR-TKIs in patients with advanced NSCLC are lacking. *Methods*: This ongoing observational study was carried out by 23 hospital-based physicians in Greece. The decision to perform cobas^®^
*EGFR* Mutation Test v2 in tissue and/or plasma at disease progression was made before enrollment. For patients with negative/inconclusive T790M plasma-based results, tissue re-biopsy could be performed. *Results*: Ninety-six (96) eligible patients were consecutively enrolled (median age: 67.8 years) between July-2017 and September-2019. Of the patients, 98% were tested upon progression using plasma and 2% using tissue/cytology biopsy. The T790M mutation was detected in 16.0% of liquid biopsies. Tissue re-biopsy was performed in 22.8% of patients with a T790M-negative plasma result. In total, the T790M positivity rate was 21.9%, not differing between patients on first- or second-generation EGFR-TKI. Higher (≥2) ECOG performance status and longer (≥10 months) time to disease progression following EGFR-TKI treatment initiation were associated with T790M positivity. *Conclusions*: Results from plasma/tissue-cytology samples in a real-world setting, yielded a T790M positivity rate lower than previous reports. Fewer than one in four patients with negative plasma-based testing underwent tissue re-biopsy, indicating the challenges in routine care settings.

## 1. Introduction

In 2018, tracheal, bronchus and lung cancer ranked as the leading cause of cancer-related deaths worldwide and in Greece, a country which ranked fourth among 185 countries in terms of age-standardized incidence rate of lung cancer (40.5 per 100,000) [1,2]. Non-small cell lung cancer (NSCLC) accounts for approximately 85% of lung cancer cases, with few patients carrying driver mutations in the gene encoding for epidermal growth factor receptor (*EGFR*) [3]. Advances in translational research have established EGFR tyrosine kinase inhibitors (TKIs) as the standard first-line therapy for NSCLC patients with sensitizing *EGFR* mutations [3]. Administration of EGFR-TKI therapy has been associated with improved outcomes and quality of life compared with doublet chemotherapy in *EGFR*-mutant NSCLC [4,5,6,7,8].

We have to take into consideration that currently, after FLAURA’s study data release, the study that compared osimertinib vs erlotinib or gefitinib in 1st line treatment, osimertinib is recommended by European Society of Medical Oncology (ESMO) guidelines, as 1st line treatment in patients expressing sensitizing mutation of *EGFR* and patient expressing T790M as de novo mutation (I, A; MBCS score v1.1 score: 4) [3,9]. During the recruitment period of this study, as osimertinib was not available as 1st line treatment, except for patients harboring the de novo T790M point mutation, there was no general consensus for a preference of any of the available first- and second-generation EGFR-TKIs in the first-line setting [3]. In fact, in the randomized phase IIB LUX-Lung 7 trial, similar overall survival (OS) but significantly better objective response rates (ORR)and progression-free survival (PFS) were reported for the second-generation EGFR-TKI afatinib versus the first-generation gefitinib [10,11]. Similarly, in the randomized phase III ARCHER 1050 trial, the second-generation TKI dacomitinib (not available in Greece) was shown to significantly improve PFS over the first-generation EGFR-TKI, gefitinib [12,13]. 

Importantly, approximately 20 to 40% of *EGFR*-mutated NSCLC patients experience primary resistance to first- or second- generation EGFR-TKIs, commonly attributed to genetic alterations, such as exon 20 insertions and the de novo T790M point mutation [14,15,16,17]. In addition, even among patients with an initial response, the majority progress after 9 to 14 months of treatment with a first- or second-generation EGFR-TKI [10,11,12,13]. Several mechanisms have been implicated in the development of acquired resistance to EGFR-TKIs, with acquisition of the T790M mutation considered the most prominent [18,19,20], detected in up to 50% of patients treated with first- or second-generation EGFR-TKIs [20,21,22,23,24]. For patients with systemic progression and a confirmed T790M mutation either with tissue biopsy or circulating-tumor DNA (ctDNA) plasma testing (and tissue re-biopsy if plasma test is negative), administration of the third generation EGFR-TKI, osimertinib, is the treatment of choice, while for patients who cannot undergo tissue biopsy or for whom a T790M mutation is not detected, the contemporary ESMO guidelines recommend switching to platinum-based chemotherapy [3,25,26].

As a result, tailoring of subsequent therapy among patients with acquired resistance to first-line EGFR-TKI requires knowledge of the T790M mutation status. Mutational analysis of *EGFR* can be performed both in tissue and liquid biopsy, with the clinical utility of the latter recognized in several clinical trials [27]. In addition to being minimally invasive and easily repeatable, the assessment of ctDNA addresses the bias stemming from the molecular heterogeneity of tumor samples and overcomes the difficulties of performing tissue re-biopsy, including, but not limited to, patient refusal, absence of a lesion amenable to re-biopsy, and safety concerns due to the patients’ declining performance status/old age [3,28,29]. While *EGFR* testing in tissue is widely available in Greece, either as a single testing or as part of a broader molecular profiling, plasma testing for detection of *EGFR* mutations in ctDNA is not yet reimbursed, resulting in access difficulties. 

In light of the above, the ‘LUNGFUL’ observational study was designed to assess the frequency of T790M acquired resistance and predictors of its presence, in patients with *EGFR*-mutated locally advanced or metastatic NSCLC who have progressed in the first-line EGFR-TKI treatment setting with first- or second-generation TKIs and have undergone molecular testing using the FDA-approved cobas^®^
*EGFR* Mutation Test v2 in tissue and/or plasma biopsy. These objectives were analyzed during the study’s interim analysis in the total study sample size, the results of which are presented herein. 

## 2. Patients and Methods

### 2.1. Study Design and Setting

This is an ongoing non-interventional, single-country, multicenter, prospective cohort study, based on primary data collection, of patients with locally advanced or metastatic *EGFR* mutation-positive NSCLC who had been treated with a first- or second- generation EGFR-TKI in the first-line setting and for whom the physicians had decided, prior to enrollment, to perform tissue and/or plasma-based liquid biopsy, using the cobas^®^
*EGFR* Mutation Test v2, at the time of disease progression in the first-line setting. 

The study is carried out by hospital-based physicians under real-world conditions of daily clinical practice. In addition, in order to represent variations in current real-world patterns of care, research sites were recruited from various geographic regions in Greece, also taking into consideration the regional setting and type of healthcare site/institution (publicly/privately owned, specialized oncology/pulmonology clinic, university clinic). The overall study duration period is approximately 36 months, including a 24-month recruitment period. The study design is presented in Figure 1.

### 2.2. EGFR Mutation Testing

All molecular pathology laboratories carrying out EGFR mutation testing applied cobas^®^ EGFR Mutation Test v2. For blood collection, cell-free DNA blood collection tubes (cfDNA BCTs; Streck, Nembraska, NE, USA) were used according to the manufacturer’s instructions, in order to ship 10 mL of whole blood at ambient temperatures (15–25 °C). cfDNA BCTs were previously shown to retain the integrity of the cfDNA and stability of blood cells preventing dilution of circulating tumor DNA (ctDNA) with wild-type genomic DNA [30]. The isolation of circulating free DNA was carried out with cobas^®^ cfDNA sample preparation kit (Roche Diagnostics, Pleasanton, CA, USA). The detection of EGFR mutations was performed with cobas^®^ EGFR Mutation Test v2 as per manufacturer’s instructions (Roche Diagnostics). Briefly, 2 mL of plasma was used to extract cfDNA using cobas^®^ cfDNA Sample Preparation Kit. The target DNA was then amplified and detected on the cobas z 480 analyser (Roche Diagnostics) using the amplification and detection reagents provided in the cobas^®^ EGFR Mutation Test v2 kit (Roche Diagnostics). Data analysis was automatically performed by EGFR Plasma Analysis Package Software version 1.0 (Roche Diagnostics) [31]. As per the analytical/non clinical evaluation of this test, a limit of detection of 100 copies/mL for T790M in plasma spiked with sheared cell line DNAs was reported [31].

Both kits are FDA approved as companion diagnostics (CoDx) in order to identify EGFR mutations (incl. T790M) in the plasma of patients with NSCLC in order to identify eligible patients for treatment with the TKI inhibitors erlotinib and osimertinib, [32,33]. As per its clinical evaluation, the cobas^®^ EGFR Mutation Test v2 for plasma, was used in the majority of clinical trials supporting osimertinib’s use in resistance T790M setting, namely AURA extension and AURA2 phase II and AURA III phase 3 trial. With cobas tissue test results as a reference, the plasma T790M positive percent agreement (PPA) was 61% and 51% by cobas plasma, in AURA II and AURA III respectively [34,35].

Disease progression could have occurred during or after discontinuation of the EGFR-TKI regimen received in the first-line setting. 

The results of the interim analysis are presented in this work, which includes analysis of data collected during the enrollment visit and the optional interim visit in the full study sample.

The study mainly involves collection of primary data, obtained prospectively during the study visits as performed per standard clinical practice or through patient self-report. Data regarding the patient’s medical and lung cancer-related history are abstracted from patient medical charts/records. Data are recorded on a web-based data capture system specifically designed for the needs of the study, which adheres to all applicable data protection regulations and requirements with regard to electronic records and database validation.

The study was designed and is being conducted in compliance with all applicable local laws and regulations, the Good Pharmacoepidemiology Practices of the International Society for Pharmacoepidemiology and the ethical principles laid down in the Declaration of Helsinki. The study was approved by the Institutional Review Boards of the participating hospital sites. All patients provided written informed consent.

### 2.3. Study Population

The study population includes outpatients, aged at least 18 years at the time of informed consent, with histologically- or cytologically-documented *EGFR* mutation-positive locally advanced or metastatic (IIIB–C/IV) NSCLC (according to the staging criteria used by the physicians in their routine practice) of any histological type who had progressed (per Investigator’s assessment) on or after first-line treatment with a first- or second- generation EGFR-TKI approved in Greece (namely erlotinib, gefitinib and afatinib), and for whom the decision to undergo tissue or plasma-based liquid biopsy, using the cobas *EGFR* Mutation Test v2, after confirmation of disease progression had been taken prior to their enrollment in the study and was separated from the physician’s decision to include the patient in this study. Prior adjuvant and neo-adjuvant chemotherapy or radiotherapy, as well as prior platinum-based chemotherapy in the context of first-line treatment for advanced/metastatic disease was allowed, provided that EGFR-TKI was administered as maintenance therapy following disease control achievement. Patients who at the time of tissue biopsy or plasma-based liquid biopsy had initiated second-line treatment were excluded from study participation. For the purposes of the study, second-line treatment was defined as switch to or addition of another agent, regardless of the drug class, including EGFR-TKI re-challenge with intervening chemotherapy. Continuation of the same EGFR-TKI, local therapy (surgery, radiotherapy) and best supportive care were not considered second-line treatment. 

### 2.4. Study Objectives 

The primary objective of the study was to assess the frequency of the T790M mutation, using the cobas *EGFR* Mutation Test v2 at the time of progression on or after first-line first- or second-generation EGFR-TKI therapy. Secondary objectives, applicable to the present analysis, were to determine the frequency of *EGFR* T790M-mediated primary and acquired (secondary) resistance to first-line EGFR-TKI therapy; to depict the patients’ *EGFR* molecular profile; to evaluate molecular testing patterns, in terms of biopsy type (tissue re-biopsy or plasma-based liquid biopsy), biopsy timing, and collection site; and to identify potential patient and clinicopathological predictive factors for T790M mutation status at the time of disease progression in the first-line setting.

### 2.5. Relevant Definitions

Index diagnosis with advanced (IIIB–IV) NSCLC was defined as the diagnosis of the disease stage present at the time of EGFR-TKI initiation in the first-line treatment setting. In case a patient had transitioned to a higher advanced disease stage in the period between the initial diagnosis and the time of initiation of EGFR-TKI in the first-line setting, index diagnosis was defined as that of diagnosis of the disease stage present at initiation of the EGFR-TKI. In regards to the definitions of primary and secondary (acquired) resistance, the following apply: primary resistance was defined as progressive disease (PD) or stable disease (of less than 6 months duration) as best response while receiving EGFR-TKI in the first-line setting, and secondary resistance was defined as progression of the disease after a period of clinical benefit; i.e., complete response (CR), partial response (PR), or durable stable disease (SD) (≥6 months). 

In addition, the progression patterns have been classified as oligoprogression, systemic progression, and central nervous system (CNS) sanctuary PD based on the following definitions: oligoprogression was considered as new sites or regrowth in a maximum of three anatomic sites; systemic progression was defined as multi-site progression, which may include both new metastatic sites as well as regrowth in previously responsive sites of disease; and CNS sanctuary progression was defined as isolated CNS failure, primarily parenchymal brain metastasis, in the absence of systemic progression [34].

### 2.6. Statistical Analysis 

Statistical analyses were performed using SAS^®^ v.9.4 (SAS Institute Inc., Cary, NC, USA). The normality of distribution of continuous variables was examined using the Shapiro-Wilk test. Summary statistics of continuous variables are presented as mean (SD) in cases data follow a normal distribution; otherwise, the median (interquartile range; IQR) is presented. Regarding proportions, 95% Clopper-Pearson confidence intervals (CIs) were calculated.

In the primary endpoint analysis, patients classified as T790M-negative based on their plasma-based liquid biopsy who were found to be T790M-positive according to the results based on the tissue re-biopsy were classified as being positive for T790M, whereas if the results based on the tissue re-biopsy were inconclusive, they were classified as being T790M-negative, according to their initial plasma-based results. The effect of selected factors of interest on the study’s primary outcome (T790M status) was examined using univariable logistic regression models. The following factors were examined: age at the time of biopsy collection upon disease progression in the first-line treatment setting (≤65 years versus >65 years); smoking status at enrollment (never-smoker versus ever-smoker), sex (female versus male), ECOG performance status (PS) at enrollment (0–1 versus ≥2), generation of first-line EGFR-TKI (second versus first), presence of exon19 deletions, L858R mutation, exon 19 deletions and/or L858R mutation, and exon 20 mutation prior to first-line treatment initiation, type of biopsy to determine T790M status (tissue versus plasma), (re)biopsy collection site (site of the primary tumor versus metastatic site), time from EGFR-TKI initiation in the first-line setting to disease progression (≥10 versus <10 months), time elapsed from first documentation of disease progression in the first-line setting to biopsy collection for *EGFR* mutation analysis (≥1 versus <1 month), best response to first-line EGFR-TKI (CR/PR/SD versus PD), and type of resistance to first-line EGFR-TKI therapy (secondary versus primary resistance). All statistical tests were two-sided and were performed at a 0.05 significance level. 

### 2.7. Sample Size

The sample size calculation was based on the study’s primary endpoint. A sample size of 115 patients (taking into consideration an estimated 15% drop out/non-evaluable rate) offers a maximum margin of error (minimum precision) of ±0.10, considering the maximum indetermination (i.e., the worst-case proportion estimate of 0.5) and a binomial two-sided confidence level of 95% using the normal approximation method. Sample size determination was performed using the statistical software package SAS v9.4 (SAS Institute, Cary, NC, USA).

## 3. Results

### 3.1. Patient Disposition and Characteristics at Enrollment

A total of 96 eligible patients were consecutively enrolled in this study by 23 hospital-based oncologists/pneumonologists between 26 July 2017 and 24 September 2019. Of these, 18 patients also attended an optional interim visit that involved tissue re-biopsy (Figure 2).

All eligible patients were Caucasians and 67.7% were females. At enrollment, the patients’ median age was 67.8 years (59.4% were aged >65 years), 42.7% of the patients were ever-smokers, and 83.3% had an ECOG performance status of 0/1 (Table 1). 

### 3.2. NSCLC Disease Characteristics at Initial NSCLC Diagnosis, at First-Line EGFR-TKI Treatment Initiation, and at the Time of Progression in the First-Line Setting

The patients’ median age at initial NSCLC diagnosis was 66.8 years, with 82 (85.4%) having been diagnosed with advanced NSCLC (including 75 diagnosed with stage IV disease). The primary tumor was adenocarcinoma in 95.8% of evaluable patients. At the time of EGFR-TKI treatment initiation in the first-line setting, nine patients (9.4%) had locally advanced and 87 (90.6%) had metastatic NSCLC, while a total of 89 patients had metastatic disease at the time of progression in the first-line setting (Table 2). 

Among patients with metastatic disease at the time of initial diagnosis, 24.0% (18/75) had >6 metastatic lesions (other than regional lymph nodes). This percentage was about the same among patients with metastatic disease at the time of EGFR-TKI treatment initiation [24.1% (21/87)], while it increased to 30.7% (27/88) among evaluable patients with metastatic disease at the time of progression in the first-line setting. The most common metastatic sites were bones, pleural effusion, contralateral lung, brain, liver, and extrathoracic lymph nodes at all examined timepoints. 

Specifically, at first-line EGFR-TKI initiation, and at disease progression in the first-line setting, 44.8% (39/87), and 50.6% (45/89) of patients with metastatic disease had metastasis to bones, 17.2% (15/87), and 23.6% (21/89) had metastasis to the liver, and 14.9% (13/87), and 30.3% (27/89), respectively, had metastasis to the brain (Figure 3).

### 3.3. NSCLC Management from Initial Diagnosis until the End of First-Line Treatment for Advanced Disease

Prior to initiation of first-line treatment with an EGFR-TKI, 38 patients had received surgical and/or pharmacological treatment and/or radiotherapy. Of those, 15 had received chemotherapy as first-line treatment for the index diagnosis prior to initiation of the EGFR-TKI (Table 3). This included six patients who initiated platinum-based combination chemotherapy while waiting for *EGFR* molecular testing results and who then received EGFR-TKI as first-line therapy, two patients who were initiated on chemotherapy while waiting for *EGFR* molecular testing results, and who received EGFR-TKI as maintenance therapy, and seven patients for whom the reason for initiating chemotherapy in the first-line setting followed by EGFR-TKI as maintenance therapy was not recorded. First-line treatment for the index advanced disease was initiated at a median (IQR) age of 66.9 (56.4–73.5) years, and included a first-generation EGFR-TKI for 39 patients (40.6%); a second-generation EGFR-TKI for 43 patients (44.8%); both a first- and a second- generation EGFR-TKI for five patients (5.2%), and receipt of a first- and/or second-generation EGFR-TKI as maintenance therapy after receipt of platinum-based combination chemotherapy as first-line therapy for nine patients (9.4%) (Table 3). 

### 3.4. Response Rates and Patterns of Disease Progression in the First-Line Treatment Setting

The ORR during first-line therapy was 51.1% (47/92) when including both confirmed and unconfirmed responses, while the respective rate including only confirmed responses was 42.6% (26/61) (Table 3). At the time of the first documented disease progression, the EGFR-TKI therapy was ongoing in 85.4% (82/96) of the patients, temporarily interrupted in 5.2% (5/96), and permanently discontinued in 9.4% (9/96) of the patients. 

The median (IQR) time to progression (i.e., time elapsed between initiation of EGFR-TKI and the first documented disease progression) was 11.0 (5.6–17.9) months in the overall population, and 11.2 (4.8–18.6) months in patients who received only a first or second generation EGFR-TKI as first-line therapy (i.e., not including receipt of EGFR-TKI as maintenance therapy), (*n* = 82). Moreover, the median (IQR) time to progression was 11.0 (3.6–17.2) in patients treated with a first-generation EGFR-TKI (*n* = 39), and 11.5 (5.8–20.1) in patients treated with a second-generation EGFR-TKI (*n* = 43).

Disease progression in the first-line setting was determined using the RECIST v1.1 criteria in 89.6% (86/96) of the patients or according to physician’s assessment based on clinical and/or imaging criteria in 10.4% (10/96) of the patients. The pattern of progression in the overall population was oligoprogression in 36.5% (35/96), systemic progression in 42.7% (41/96), CNS sanctuary progression in 13.5% (13/96), and clinical progression in the absence of any radiologic evidence of progression (i.e., clinical deterioration of disease-related symptoms) in 7.3% (7/96) of the patients (Figure 4). 

### 3.5. EGFR Gene Mutation Profile Prior to EGFR-TKI Treatment Initiation and at Disease Progression in the First-Line Setting

Prior to EGFR-TKI therapy initiation, exon 19 deletions were detected in 59.4% of the patients and the L858R mutation in 27.1%; the frequencies of other mutations are presented in Table 2. De novo T790M mutation was identified in 2.1% of the patients.

At the time of disease progression in the first-line setting, *EGFR* molecular testing was performed with the cobas^®^
*EGFR* Mutation test v2 using plasma-based liquid biopsy in 94 (97.9%) patients, and a tissue/cytology sample in the remaining two patients (2.1%; one sample was tissue from the site of the primary tumor and one was pleural effusion). The results of the test were conclusive/valid for all patients during this initial testing (Figure 2). At the time of the first biopsy collection, in plasma or tissue, first-line EGFR-TKI therapy was ongoing in 56.3%, discontinued in 38.5%, and temporarily interrupted in 5.2% of the patients. 

The T790M mutation was detected in 16.0% (15/94) of plasma biopsies and in one tissue biopsy from the primary tumor; a pleural effusion specimen was negative for T790M. Of the patients found to be negative for T790M (Figure 2), re-biopsy was not planned for the patient for whom molecular testing was performed using a pleural effusion specimen, and for 57 of the 79 patients found negative for T790M by plasma biopsy. Of the remaining 22 patients for whom re-biopsy and molecular testing with cobas^®^
*EGFR* Mutation test v2 was planned, re-biopsy was actually performed in 18 patients (22.8%); re-biopsy involved a tissue sample in 16 cases (from the site of the primary tumor in nine cases and from a metastatic site in seven cases) and cerebrospinal fluid in two cases. The results were conclusive/valid in 13 samples. The T790M mutation was identified in five of the samples, two from a primary tumor site and three from metastatic sites (two from lymph nodes and one from lung metastasis). Thus, taking into consideration the results of re-biopsy a total of 21 patients were found to be positive for the T790M-mutation, and the remaining 75 patients were classified as being T790M-negative (Figure 2). The respective T790M-positive and T790M-negative rates were 21.9% (95% CI: 14.1–31.5) and 78.1% (95% CI: 68.5–85.9).

Among patients who underwent plasma-based molecular testing, exon 19 deletions were identified in 29.8% (28/94), the L858R mutation in 16.0% (15/94), and the T790M mutation in 16.0% (15/94) of the patients. Among the two patients whose initial molecular testing was performed in a tissue sample, exon 19 deletions were identified in 100.0% (2/2) and the T790M mutation in 50.0% (1/2). In the overall population, according to both the initial biopsy and the re-biopsy results, the frequencies of exon 19 deletions, the L858R, and the T790M mutations were 30.0% (30/96), 16.7% (16/96), and 21.9% (21/96), respectively. The patterns of the *EGFR* mutations identified during the initial cobas *EGFR* Mutation test v2 using plasma biopsy (*n* = 94), during the initial testing with tissue biopsy (*n* = 2), and in the overall population taking into consideration the conclusive/valid results of molecular testing during re-biopsy and the results of the initial biopsy for all other patients are presented in Figure 5 (panels A, C and E, respectively). Among the 21 T790M-positive patients, the most frequently co-occurring mutations were exon 19 deletions (61.9%; *n* = 13), L858R (28.6%; *n* = 6), and exon 20 insertions (9.5%; *n* = 2). Interestingly, lower rates of mutations were identified among the 75 T790M-negative patients, with exon 19 deletions again being the most frequently identified (22.7%; *n* = 17), followed by L858R (13.3%; *n* = 10) and G719X (4.0%; *n* = 3). The patterns of co-occurring mutations at initial molecular testing using a plasma specimen, at initial testing using a tissue/cytology specimen, and overall using both the molecular test results of the initial biopsy and those of the re-biopsy are displayed in Figure 5 (panels B, D, and F). 

The median (IQR) time from first documentation of disease progression to collection of the biopsy specimen with a positive T790M result (whether this was the initial plasma/tissue/cytology biopsy or the tissue/cytology re-biopsy sample), or to the initial biopsy with a negative result (in cases that a re-biopsy was not performed or that the re-biopsy confirmed the negative results) was 0.6 (0.2–1.4) months.

The mutation patterns prior to initiation of EGFR-TKI in the first-line treatment setting and their shift at the time of progression in the first-line setting among patients identified to be T790M-positive at progression are displayed in Table 4. For 18 of the 21 patients the only identified change included acquisition of the T790M. 

### 3.6. T790M-Mediated Primary and Secondary Resistance Rates to First-Line EGFR-TKI and Patterns of Progression in T790M-Positive and T790M-Negative Patients

The EGFR-T790M mediated primary and secondary resistance rates were 8.0% (2/25) and 28.4% (19/67) when examining both confirmed and unconfirmed responses, while they were 4.5% (1/22) and 33.3% (13/39) when examining only confirmed responses. The patterns of progression in the first-line treatment setting among the T790M-positive and T790M-negative subpopulations are presented in Figure 4. 

### 3.7. Association of Patient and Clinicopathological Characteristics with EGFR-T790M Status 

Patient and clinicopathological characteristics in T790M-positive and T790M-negative patients presented in Table 5 were examined as to their association with T790M mutation status (Table 5). Of the examined factors, patients with ECOG performance status of 0–1 at enrollment compared to those with ECOG performance status >2 were less likely to be T790M-positive (*p* = 0.026), while patients with a longer time to disease progression following initiation of EGFR-TKI were more likely to be T790M-positive (*p* = 0.027) (Table 5). 

### 3.8. Deaths and Study Withdrawal

Until the data cut-off date in the context of this interim analysis, 20 (20.9%) patients had died. In particular, 14 deaths occurred prior to initiation of second-line treatment and six after initiation of second-line treatment in the absence of documented disease progression in the second-line setting. In addition, 18 patients (18.8%) were withdrawn from the study due to disease progression in the second-line setting, and one patient was lost-to-follow-up. Therefore, in total 39 patients (40.6%) had discontinued study participation by the interim data cut-off date.

## 4. Discussion

The ‘LUNGFUL’ study provides real-world evidence from routine care clinical settings in Greece on the frequency of *EGFR* mutations, focusing on T790M, in patients with *EGFR*-mutated advanced NSCLC who have progressed in the first-line EGFR-TKI (1st or 2nd generation) treatment setting. In this study, molecular testing was performed in tissue and/or plasma biopsy samples using the cobas^®^
*EGFR* Mutation Test v2. 

Prior to initiation of EGFR-TKI therapy, exon 19 deletions and the L858R substitution were the most frequently detected mutations, while the de novo T790M substitution was detected in 2.1% of the patients with *EGFR* mutations, in agreement with the range (1.0–2.2%) reported by other studies [35,36,37], including a retrospective study by the Hellenic Co-operative Oncology Group [38]. At disease progression on or after first-line EGFR-TKI therapy, 21.9% of the patients were T790M-positive. The rate observed in ‘LUNGFUL’ is lower than that reported in other international studies [20,21,39,40,41,42], but is higher than the 16% rate obtained by cobas in liquid biopsies in another study from Greece, which reported results both from next generation sequencing and from cobas analysis [43]. T790M positivity rate in our study was similar to that of a recent study performed in Korea, with positivity rate of 23%, but with the main difference that this study was conducted in Asian population [44]. The incidence of T790M among NSCLC patients who progressed upon EGFR-TKI as first-line therapy reported by other studies ranges from 36% to 70%, not largely differing between patients who have previously received a first- or second-generation EGFR-TKI [20,21,24,39,40,42,43]. Although the observed differences could reflect real variation among the populations analyzed, the applied method of molecular testing could also account for some of the discrepancies, as the sensitivity and specificity of the methods vary [45]. Nonetheless, in a combined analysis of AURA Extension and AURA2, which similarly to the present study utilized cobas^®^ methodology for detection of *EGFR* mutations, the pooled T790M rate was 63%, which is much higher than that in our study [46]. It is noted that in this pooled analysis, the rate of T790M positivity did not differ between Asian and non-Asian patients, which suggests that the source of the variation between this study and ours is likely not the difference in ethnic background, and could at least partly be attributable to different tumor burden and number of prior treatment lines received by the patients, as all patients in our study had received EGFR-TKI in the first-line setting, whereas in the AURA trials patients could have received more than one prior line of therapy [46]. On the other hand, both in ‘LUNGFUL’ and in the pooled analysis of AURA extension and AURA2, the generation of the immediately prior EGFR-TKI was not shown to be associated with the T790M rate [46]. 

At disease progression in the first-line setting, the co-occurrence of T790M mutation with exon 19 deletion in our study was almost 2.2-fold higher than the co-occurrence of T790M with L858R (61.9 versus 28.6%). A higher proportion of co-occurring exon 19 deletions than L858R mutation in T790M-positive patients has been reported elsewhere [42,43], with the respective rates being 67.5% and 29.5% in the pooled analysis of the AURA extension and AURA2 trials [46]. Importantly, in our study, the rate of detection of exon 19 deletions was higher than that of the L858R substitution in T790M-positive and T790M-negative patients alike. This finding is similar to that presented in a 2019 Hellenic national conference from a series of 403 unique plasma samples examined by cobas testing (I. Boukovinas unpublished data from the 11th Training Seminar in Clinical Oncology, held in November 2019). In particular, the T790M detection rate in these 403 samples was 14%. Moreover, exon 19 deletions were detected in 68.5% of T790M-positve and in 19.5% of T790M-negative patients, while L858R mutation was detected in 7.4% and 6.1% of T790M-positive and T790M-negative patients, respectively. A higher frequency of exon 19 deletions compared to the L858R mutation among Caucasian populations has been consistently reported [46]. 

It also of interest to highlight that although 95.2% (20/21) of T790M positive patients had co-occurring mutations, only 41.3% (31/75) of T790M negative patients carried activating mutations. Among other reasons, this finding may be explained by a potential higher rate of incomplete tumor ctDNA shedding among T790M-negative patients, a mechanism also accounting for false negative T790M results [47].

Furthermore, previous studies have also reported a higher rate of T790M positivity in patients with initial exon 19 deletion (range: 38–63%), than in patients with pre-existing L858R mutations (range: 26–43%) [42,48]. In ‘LUNGFUL’, the frequencies of T790M were about 23.0% both in patients with exon 19 deletions and in those with the L858R mutation prior to initiation of EGFR-TKI therapy, with neither of the two factors shown to predict T790M status. Conversely, a significant association was observed between T790M status and type of biopsy. Previous studies comparing the prevalence of T790M between cytology and tissue samples have revealed no significant differences [42,48]. Regarding the concordance rates between tumor tissue and plasma using cobas, rates of 64% to 79% have been reported [49,50]. In our patient population, of the 13 cases with negative T790M results using a plasma biopsy, 8 (61.5%) were also shown to be negative using a tissue sample, while the remaining five cases were found to be positive. 

Of note, in 46 samples no mutations were detected after progression, a number relatively high. As shown in a recent exploratory ctDNA analysis from patients enrolled in the AURA 3 study, almost 25% of patients treated with osimertinib were defined as non-shedders as they were not found positive for any of the three main EGFR mutations (T790M, del19, L858R) [51]. While there were no clear associations of the EGFR tumor shedding status with sex, race, smoking status and performance status according to Eastern Cooperative Oncology Group (ECOG), an association with the baseline tumor target lesion size has been reported. In addition it has been previously reported that the detection of EGFR mutations in both plasma and tissue samples was associated with metastatic status [49]. More specifically, the presence of extrathoracic disease(M1b) was associated with the detection of EGFR mutations in plasma with statistical significance when compared to intrathoracic (M1a/M0) disease. In the above context and taking into consideration that i. all labs participating in the LUNGFUL trial for the detection of T790M in plasma, using cobas EGFR Mutation Test v.2 have been successfully participated in a robin trial for the detection of different EGFR mutations in different concentrations in plasma [52] and that ii. cfDNA BCTs were used to retain both the integrity of the cfDNA population and the stability of blood cells preventing dilution of ctDNA with wild-type genomic DNA, preanalytic and analytic issues may not account for the inability to identify any mutation in the plasma of 46 patients. 

Rather this should be attributed to either “non shedder” status or disease burden status as explained above. It is generally accepted that plasma-based biopsy helps circumvent some of the challenges of re-biopsy, stemming from procedure invasiveness and heterogeneity of the tumor tissue [28,29,45]. In fact in our study, while EGFR mutation testing at the time of diagnosis in the context of first-line treatment initiation in the advanced setting had been performed using a sample from tissue biopsy in more than 80% of the patients, this proportion at the time of re-biopsy on or after progression in the first-line treatment setting was only 1%. Moreover, of the patients who were negative for T790M based on liquid biopsy, and for whom guidelines [3] recommend a tissue re-biopsy, such a biopsy was not planned for 72%, with patient unwillingness, poor clinical status/old age, and difficulties with specimen acquisition being the main reasons. Similarly, in a study conducted in China, the rate of non-performance of re-biopsy at the time of progression after receipt of a first-generation EGFR-TKI was rather high (about 47%). The main reasons for not performing re-biopsy included lesion sizes and/or locations unsuitable for biopsy, poor health, older age or severe comorbidity, and patient unwillingness [53]. These constraints of tissue re-biopsy may reflect logistic and infrastructure hurdles in hospitals and merit further actions from the healthcare community, as plasma biopsies can give false-negative results, so patients who could potentially benefit from treatment with third-generation TKIs might be missed [54]. Based on findings from AURA2 and AURA3 studies, specificity is 79% for the T790M, whereas it is not an issue for the activating EGFR mutations. However, liquid biopsy is a valid option to consider when re-biopsy is not an option for the treating physicians [55].

With respect to disease progression, compared to T790M-negative patients, T790M-positive patients exhibited higher oligoprogression rates (42.9% versus 34.7%) and lower CNS sanctuary progression rates (4.8% versus 16.0%), while rates of systemic progression (42.9% versus 42.7%) were about the same. In a previous report, the T790M positive rate was significantly higher in patients with local progression compared to those with gradual progression and dramatic progression [41]. Moreover in ‘LUNGFUL’, the median time to progression since initiation of EGFR-TKI in the first-line setting was 11.0 months in the overall population, being similar among patients treated with only a first-generation and only a second-generation EGFR-TKI (11.0 versus 11.5 months), and aligning with findings from the phase III trials of first and second generation EGFR-TKIs [4,5,6,7,8,11,12,13,23]. Logistic regression analysis revealed a significant association of T790M positive mutation status with the time elapsed between EGFR-TKI initiation in the first-line setting and the first documented disease progression. This observation is similar to the finding of a previous study that patients with longer duration of EGFR-TKI treatment had a higher T790M positive rate [56], but differs from other studies, which showed no such association [57,58]. Moreover, a longer period of EGFR-TKI treatment continuation post disease progression has also been shown to be associated with a higher rate of T790M positivity [57]. Identifying predictors of T790M positivity post-progression in the first-line setting is a field of active research as treatment-decision making in the second-line setting largely depends on T790M status. The results of our final analysis will elaborate on the treatment management strategies employed in the second-line setting depending on the patients’ patterns of progression, prior therapy, patterns of metastasis, and T790M mutation status. 

The main limitations of this study are attributed to its observational design and primarily include inherent patient selection and information bias, the latter of which is considered to be small due to the low missing data rates. In order to mitigate potential patient selection bias, physicians were requested to consecutively enroll the first eligible patients (based on the site-specific target) attending their clinic over the pre-specified study recruitment period. In addition, the Investigators’ decision to perform *EGFR* molecular testing post-disease progression using the cobas^®^
*EGFR* Mutation Test v2 was based on current medical practice and preceded the consideration of the patient’s eligibility for enrollment into the study. Moreover, there may be information bias regarding *EGFR* mutation testing at initial diagnosis, at both an inter- and intra-patient level, since testing was performed at several independent laboratories, employing different assays according to routine practice. Thus, any inferences that may be drawn regarding changes in the *EGFR* mutational profile between the start EGFR-TKI treatment and the time of progression in the first-line treatment setting should take into consideration this source of potential variability. This source of bias does not apply for testing performed at the time or after the first PD, since a uniform test approved by the FDA was used, pre-analytical factors were the same and a pilot plasma-ctDNA ring trial for the cobas^®^ EGFR Mutation Test in clinical diagnostic laboratories performed in 2018 in Greece, showed that genotyping results were satisfactory [52]. Furthermore, while a total of 96 patients were evaluable for the study’s primary objective meeting the planned study size, a very high proportion of patients (84%) -identified to be negative by molecular testing performed in a plasma sample-did not undergo re-biopsy or had inconclusive results. This may have resulted in an underestimation of the study’s primary objective pertaining to the frequency of T790M positivity, as supported by the fact that among patients found T790M-negative based on plasma biopsies who underwent tissue re-biopsy, the percentage of T790M positivity was as high as 38%. It is noted that it was within the study’s aim to record the re-testing frequency in the routine care in Greece.

This study was designed in 2016, when osimertinib was only approved by EMA as a 2nd line treatment for patients harboring the resistance mutation T790M or as a 1st line treatment for patients expressing T790M as de novo mutation. At the time being, after FLAURA’s study data release, osimertinib is recommended by ESMO guidelines, as 1st line treatment in patients expressing sensitizing mutation of EGFR and patient expressing T790M as de novo mutation with the score [I, A; MBCS score v1.1 score: 4]. This treatment algorithm evolvement does not affect the primary objective of our study, which is to assess the frequency of T790M mutation in the real-world Greek population, regarding patients receiving 1st line treatment with 1st or 2nd generation TKIs [3,9].

Regarding external validity, the study population was enrolled from sites located in five of the 13 administrative regions of Greece, which are home to 67% of the overall Greek population, aiding the geographic diversity of the sites and generalizability of the findings. Representativeness was also facilitated by the enrollment of patients by 23 physicians treating patients with lung cancer in the primary care hospital outpatient or private practice setting, accounting for variations in medical practice paradigms.

## 5. Conclusions

In the real-world clinical setting in Greece, a 21.9% T790M positivity rate was detected, based on cobas^®^ molecular testing in plasma and/or tissue biopsy at the time of progression in the first-line setting with first- and second-generation EGFR-TKIs. The rate of testing in tissue/cytology samples was very low, with only 15 results based on tissue/cytology samples. The T790M positive rate was lower based on results from plasma (16.0%) than from tissue/cytology (40.0%) biopsies. The overall rate (which mainly reflects the rate in plasma biopsies) is lower compared to previous reports. T790M positivity was shown to be similar among patients who had previously received a first- or second-generation EGFR-TKI, while it was higher in those with a longer period between treatment initiation and disease progression in the first-line setting. Presence of exon 19 deletions and L858R mutations prior to initiation of first-line treatment did not predict acquisition of T790M. Upon disease progression in the first-line setting, re-biopsy included a plasma sample in nearly all patients, while among those who tested negative in their plasma sample, fewer than one in four underwent a tissue re-biopsy. This underscores the challenges of performing tissue re-biopsy in routine care settings, which can lead to patients not considered eligible for certain therapies from which they can benefit, and merits further actions from the healthcare community.

## Figures and Tables

**Figure 1 cancers-13-03172-f001:**
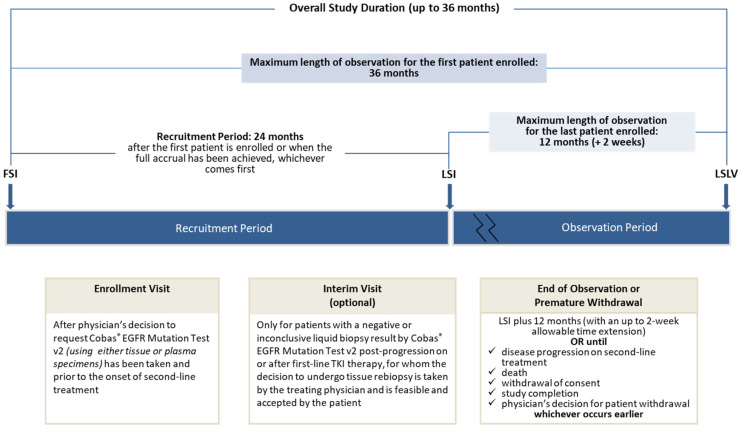
Overview of the study design. FSI: First Subject In; LSI: Last Subject In; LSLV: Last Subject Last Visit.

**Figure 2 cancers-13-03172-f002:**
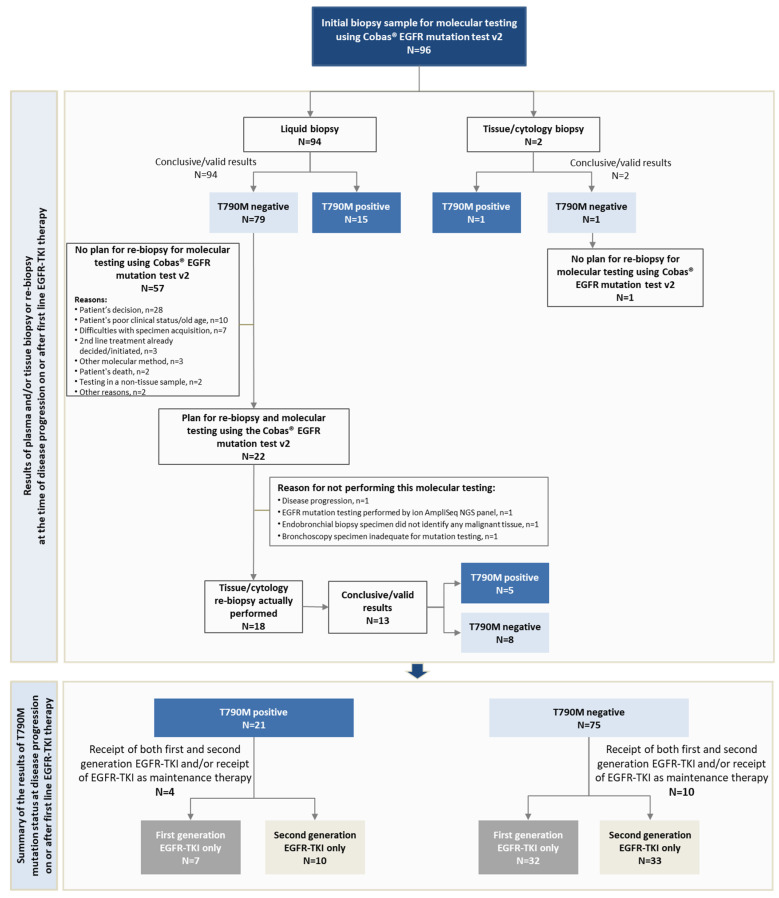
Biopsy and re-biopsy workflow and distribution of patients by T790M status and generation of the EGFR-TKI therapy received in the first-line treatment setting.

**Figure 3 cancers-13-03172-f003:**
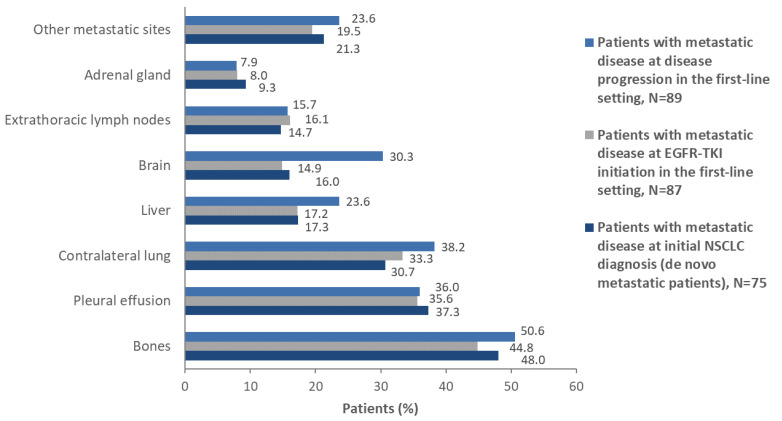
Sites of metastasis at initial diagnosis, EGFR-TKI initiation in the first-line setting, and at disease progression in the first-line setting, in patients of the overall population with metastatic disease.

**Figure 4 cancers-13-03172-f004:**
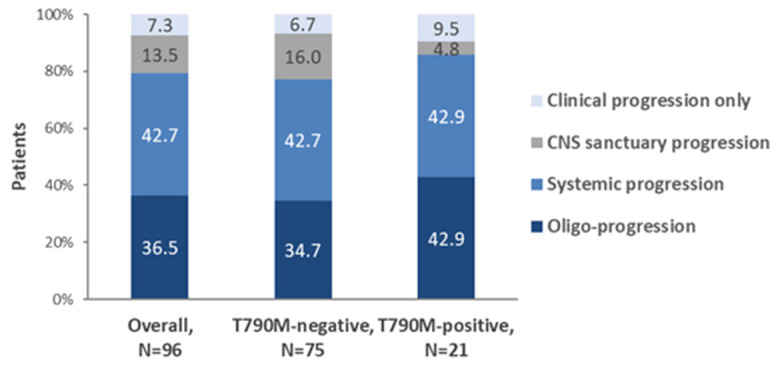
Patterns of disease progression in the first-line treatment setting among the overall population and the T790M-positive and negative subpopulations.

**Figure 5 cancers-13-03172-f005:**
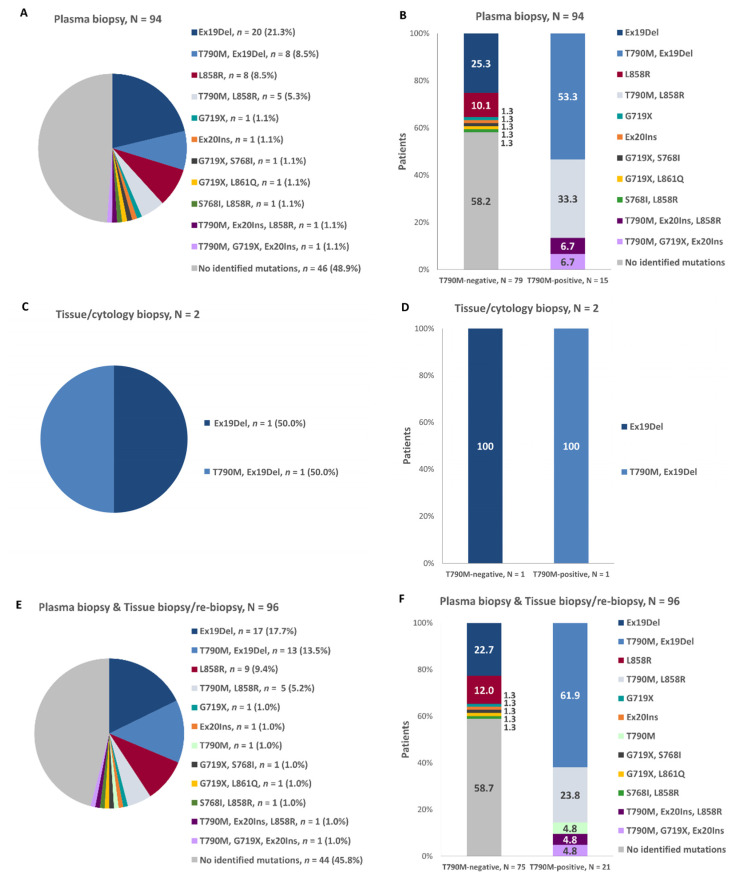
Mutations identified by cobas^®^
*EGFR* Mutation Test v2 during the initial plasma biopsy (**A**,**B**), initial tissue/cytology biopsy (**C**,**D**) and the initial plasma biopsy and the initial tissue biopsy or the re-biopsy (**E**,**F**), in the overall population (**A**,**C**,**E**) and the T790M-positive and–negative subpopulations (**B**,**D**,**F**). Ex19Del: exon 19 deletion; Ex20Ins: exon 20 insertion.

**Table 1 cancers-13-03172-t001:** Patient characteristics at enrollment.

Patient Enrollment Characteristics	Value (Mean or Median or %)
Age (*N* = 96), median (IQR), years	67.8 (57.6–74.9)
≤65 years, *n* (%)	39 (40.6)
>65 and ≤75 years, *n* (%)	33 (34.4)
>75 years, *n* (%)	24 (25.0)
Females (*N* = 96), *n* (%)	65 (67.7)
Caucasian (*N* = 96), *n* (%)	96 (100.0)
BMI (*N* = 77), mean (SD), kg/m^2^	26.3 (4.8)
Obese (BMI ≥ 30 kg/m^2^) (*N* = 77), *n* (%)	17 (22.1)
Smoking status (*N* = 96)	
Former smokers, *n* (%)	33 (34.4)
*Pack-years, median (IQR)*	16.0 (10.0–40.0)
Current smokers, *n* (%)	8 (8.3)
*Pack-years, median (IQR)*	17.5 (11.3–47.5)
ECOG performance status (*N* = 96), *n* (%)	
0	51 (53.1)
1	29 (30.2)
2	12 (12.5)
3	4 (4.2)
Medical/Surgical history and comorbidities (excluding surgeries for NSCLC) (*N* = 96), *n* (%)	51 (53.1)
Comorbidities (*N* = 96), *n* (%)	43 (44.8)
Hypertension, *n* (%)	21 (21.9)
Hypothyroidism, *n* (%)	11 (11.5)
Diabetes mellitus, *n* (%)	10 (10.4)
Dyslipidemia, *n* (%)	10 (10.4)

BMI: body mass index, IQR: interquartile range, NSCLC: non-small cell lung cancer, SD: standard deviation.

**Table 2 cancers-13-03172-t002:** NSCLC characteristics at initial and index NSCLC diagnosis and at initiation of EGFR-TKI in the first-line setting.

Age at initial diagnosis (*N* = 96), median (IQR), years	66.8 (55.5–73.4)
Specimen used for documentation/confirmation of the initial diagnosis (*N* = 96), *n* (%)	
Histological	69 (71.9)
Cytological	14 (14.6)
Histological and cytological	13 (13.5)
NSCLC stage at initial disease diagnosis (*N* = 96), *n* (%)	
I: IB	1 (1.0): 1 (1.0)
II: IIA, IIB	5 (5.2): 4 (4.2), 1 (1.0)
III: IIIA, IIIB, IIIC, Locally advanced (unspecified stage)	15 (15.6): 8 (8.3), 5 (5.1), 1 (1.0), 1 (1.0)
IV	75 (78.1)
Primary tumor histological classification (*N* = 95), *n* (%)	
Adenocarcinoma	91 (95.8)
Squamous cell carcinoma	3 (3.2)
Adenosquamous carcinoma, squamous cell carcinoma-predominant	1 (1.1)
Age at index diagnosis with advanced NSCLC (*N* = 96), median (IQR), years	66.8 (56.3–73.4)
Age at EGFR-TKI treatment initiation (*N* = 96), median (IQR), years	66.9 (56.4–73.5)
NSCLC stage at index diagnosis (*N* = 96), *n* (%)	
III: IIIB, IIIC, Locally advanced (unspecified stage)	9 (9.4): 6 (6.3), 1 (1.0), 2 (2.1)
IV	87 (90.6)
Criteria used for NSCLC staging at index diagnosis (*N* = 96), *n* (%)	
AJCC/UICC 6th edition	1 (1.0%)
AJCC/UICC 7th edition	35 (36.5%)
AJCC/UICC 8th edition	57 (59.4%)
Unknown	3 (3.1%)
Assays used for *EGFR* mutation testing prior to initiation of first-line EGFR-TKI (*N* = 96), *n* (%)	
cobas^®^ *EGFR* Mutation Test v2	37 (38.5)
Other cobas^®^ tests	12 (12.5)
Next-generation sequencing	28 (29.2)
Other assays	15 (15.6)
Unspecified assay	4 (4.2)
Sample used for *EGFR* mutation testing prior to initiation of first-line EGFR-TKI (*N* = 96), *n* (%)	
Tumor tissue	80 (83.3)
Cytology sample	9 (9.4)
Plasma	7 (7.3)
*EGFR* mutations identified prior to initiation of first-line EGFR-TKI (*N* = 96), *n* (%)	
Exon 19 deletion	56 (58.3)
L858R	26 (27.1)
Exon 20 insertion	6 (6.3)
G719X	5 (5.2)
S768I	3 (3.1)
L861Q	2 (2.1)
T790M	2 (2.1)
Other mutations (E709-T710>D, R776S, R836C, V765M)	4 (4.2)
Exon 19 unspecified mutation	1 (1.0)

Index diagnosis is defined as diagnosis with stage IIIB–IV disease in the context of EGFR-TKI initiation in the first-line setting. AJCC/UICC: American Joint Committee on Cancer/Union for International Cancer Control, EGFR-TKI: Epidermal growth factor receptor-tyrosine kinase inhibitor, IQR: interquartile range, NSCLC: non-small cell lung cancer.

**Table 3 cancers-13-03172-t003:** NSCLC therapeutic management in the first-line treatment setting and response to treatment.

First-Line Treatment for the Index NSCLC Diagnosis (*N* = 96)		*n* (%)
First generation EGFR-TKI without preceding chemotherapy		35 (36.5)
First generation EGFR-TKI with preceding chemotherapy		4 (4.2)
Second generation EGFR-TKI without preceding chemotherapy		41 (42.7)
Second generation EGFR-TKI with preceding chemotherapy		2 (2.1)
Both a first- and a second-generation EGFR-TKI		5 (5.2)
First- and/or second-generation EGFR-TKI as maintenance therapy after receipt of platinum-based chemotherapy		9 (9.4)
Best response in the first-line treatment setting, *n* (%)		
	Confirmed and not confirmed responses (*N* = 92)	Only confirmed responses (*N* = 61)
Complete response	7 (7.6)	5 (8.2)
Partial response	40 (43.5)	21 (34.4)
Stable disease	25 (27.2)	15 (24.6)
Progressive disease	20 (21.7)	20 (32.8)
Primary and secondary resistance in the first-line treatment setting, *n* (%)		
	Confirmed and not confirmed responses (*N* = 92)	Only confirmed responses (*N* = 61)
Primary resistance	25 (27.2)	22 (36.1)
Secondary resistance	67 (72.8)	39 (63.9)

EGFR-TKI: Epidermal growth factor receptor-tyrosine kinase inhibitor, NSCLC: non-small cell lung cancer.

**Table 4 cancers-13-03172-t004:** *EGFR* mutation profile of T790M positive patients at the time of progression on or after first-line EGFR-TKI prior to first-line EGFR-TKI initiation and at disease progression on or after first-line EGFR-TKI therapy.

Prior to First-Line EGFR-TKI Initiation	at Disease Progression on or after First-Line EGFR-TKI
Exon 19 deletion, *n* = 13	T790M, Exon 19 deletion, *n* = 12
	T790M, *n* = 1
L858R, *n* = 5	T790M, L858R, *n* = 5
G719X, *n* = 1	T790M, G719X, Exon 20 insertion, *n* = 1
Exon 20 insertion, L858R, *n* = 1	T790M, Exon 20 insertion, L858R, *n* = 1
T790M, Exon19 unspecified mutation, *n* = 1	T790M, Exon 19 deletion, *n* = 1

EGFR-TKI: Epidermal growth factor receptor-tyrosine kinase inhibitor.

**Table 5 cancers-13-03172-t005:** Patient and clinicopathological characteristics according to the T790M mutation status.

Patient Characteristics		T790M-Positive	T790M-Negative	
		*n* (%)	*n* (%)	*p*-Value
Age at the time of biopsy collection upon disease progression on or after first-line EGFR-TKI treatment (*N* = 96)	>65 years	13 (61.9)	44 (58.7)	0.790
	≤65 years	8 (38.1)	31 (41.3)	
Smoking status at enrollment (*N* = 96)	Ever-smoker	10 (47.6)	31 (41.3)	0.607
	Never-smoker	11 (52.4)	44 (58.7)	
Sex (*N* = 96)	Male	5 (23.8)	26 (34.7)	0.350
	Female	16 (76.2)	49 (65.3)	
ECOG performance status at enrollment (*N* = 96)	≥2	7 (33.3)	9 (12.0)	0.026
	0–1	14 (66.7)	66 (88.0)	
Generation of first-line EGFRI-TKI (*N* = 82) ^†^	First-generation	7 (41.2)	32 (49.2)	0.555
	Second-generation	10 (58.8)	33 (50.8)	
Exon 19 deletion prior to initiation of first-line EGFR-TKI (*N* = 96)	No	8 (38.1)	32 (42.7)	0.707
	Yes	13 (61.9)	43 (57.3)	
L858R mutation prior to initiation of first-line EGFR-TKI (*N* = 96)	No	15 (71.4)	55 (73.3)	0.862
	Yes	6 (28.6)	20 (26.7)	
Exon 19 deletion and/or L858R mutation prior to initiation of first-line EGFR-TKI (*N* = 96)	No	2 (9.5)	12 (16.0)	0.463
	Yes	19 (90.5)	63 (84.0)	
Exon 20 insertion prior to initiation of first-line EGFR-TKI (*N* = 96)	No	19 (90.5)	65 (86.7)	0.643
	Yes	2 (9.5)	10 (13.3)	
Type of biopsy to determine T790M status (*N* = 96)	Plasma-based liquid	15 (71.4)	74 (98.7)	0.002
	Tissue	6 (28.6)	1 (1.3)	
Tissue biopsy/re-biopsy collection site (*N* = 17)	Metastatic site	3 (50.0)	4 (36.4)	0.587
	Site of the primary tumor	3 (50.0)	7 (63.6)	
Time elapsed between EGFR-TKI initiation in the first-line setting and first documented disease progression (*N* = 82) ^†^	<10 months	3 (17.6)	32 (49.2)	0.027
	≥10 months	14 (82.4)	33 (50.8)	
Time from first documentation of disease progression in the first-line setting to biopsy collection for *EGFR* mutation analysis with the cobas^®^ *EGFR* Mutation Test v2 (*N* = 96)	<1 month	11 (52.4)	53 (70.7)	0.121
	≥1 month	10 (47.6)	22 (29.3)	
Best response to first-line EGFR-TKI based on confirmed and not confirmed responses (*N* = 92)	PD	1 (4.8)	19 (26.8)	0.060
	CR/PR/SD	20 (95.2)	52 (73.2)	
Type of resistance to first-line EGFR-TKI based on confirmed and not confirmed responses (*N* = 92)	Primary resistance	2 (9.5)	23 (32.4)	0.054
	Secondary resistance	19 (90.5)	48 (67.6)	

^†^ Analysis excludes patients who, in the first-line treatment setting, had received both a first- and a second-generation EGFR-TKI, as well as those who had received EGFR-TKIs as maintenance therapy. All *p*-values were derived from univariable logistic regression analyses. EGFR-TKI: Epidermal growth factor receptor-tyrosine kinase inhibitor.

## Data Availability

The data presented in this study are contained within the article. The data are not publicly available due to restrictions that apply to the availability of the data (e.g., privacy or ethical). Datasets from this study may be available upon request from the corresponding author and provided upon approval from the sponsor and in accordance with data privacy and ethical provisions.

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
