# Peer review of "(untitled)"

_cancers, 2021, doi:10.3390/cancers13133172_

Round 1

Reviewer 1 Report

The Authors presented the results of an interim analysis of a real-world, observational ongoing study to assess the molecular epidemiology of EGFR resistance mutations  after  progression  with  first-  or  second-generation  EGFR-TKIs  in  patients  with  advanced NSCLC. They found that T790M mutation was detected in 16.0% of liquid biopsies. Moreover, tissue re-biopsy was performed in only 22.8% of patients with a T790M-negative plasma result. In total, the T790M positivity rate was 21.9%, much lower than than reported in literature. Data on percentage of T790M positivity after EGFR TKIs are well known in literature and these results did not add new information: the lower percentage of T790M positivity can be explained with the low rate of tissue re-biopsy. Moreover, the new standard first-line treatment of patients with EGFR mutations (with osimertinib) further reduce the applicability and originality of these data.

Author Response

Dear Mme/Sir

We are grateful for giving us the opportunity to revise our manuscript. We are grateful to all reviewers for their time and efforts  to improve the quality of our manuscript. In the revised version of our manuscript we have incorporated all suggested  comments from the reviewers. Please find below a point by point reply to the  comments: 

REVIEWER 1

The Authors presented the results of an interim analysis of a real-world, observational ongoing study to assess the molecular epidemiology of EGFR resistance mutations  after  progression  with  first-  or  second-generation  EGFR-TKIs  in  patients  with  advanced NSCLC. They found that T790M mutation was detected in 16.0% of liquid biopsies. Moreover, tissue re-biopsy was performed in only 22.8% of patients with a T790M-negative plasma result. In total, the T790M positivity rate was 21.9%, much lower than than reported in literature. Data on percentage of T790M positivity after EGFR TKIs are well known in literature and these results did not add new information: the lower percentage of T790M positivity can be explained with the low rate of tissue re-biopsy. Moreover, the new standard first-line treatment of patients with EGFR mutations (with osimertinib) further reduce the applicability and originality of these data.

Response:

We really appreciate these constructive comments. The LUNGFUL study was not  designed with the intention to offer new data regarding the percentage of appearance of T790M mutation, after 1st line treatment with 1st or 2nd generation TKIs, since this data is well present in the literature for some  years  now. As we point out in the introduction and discussion sessions, the LUNGFUL study is a real-life, prospective study that provides further evidence on the daily clinical practice,   outside the context of a clinical trial. In fact, the aim of the study was to identify potential discrepancies between data from clinical trials and real-world data, and to provide possible explanations for these discrepancies.  Another important point of this study is that the EGFR testing (plasma and tissue) was performed with the same methodology in all patients (Cobas vs2), reducing biases and methodological errors. The low re-biopsy rate in negative EGFR plasma test reflects reality, and as a result impacts  the final positivity rate, decreasing the chance of patients to be treated with the optimal therapeutic option in 2nd line. Finally, as real-world evidence studies add important information are important adjuncts to clinical trials and are requested and accepted by the health community including  Health Regulatory Authorities (Health Technology  Assessment, negotiation committees etc.).

All changes have been highlighted in the Revised manuscript for your convenience

Reviewer 2 Report

This is a prospective study using plasma samples to detect the secondary EGFR-T790M mutation in advanced NSCLC patients with acquired resistance to first- and secondary-generations of EGFR-TKI. The EGFR-T790M positive rate in this study was 21.8% which was relative lower than that in previous studies.  A recent study  (Transl Lung Cancer Res. 2021 Feb;10(2):878-888.) showed similar results to this study and concluded that the sensitivity of plasma EGFR test for secondary T790M mutation in NSCLC with acquired resistance to EGFR-TKIs was low and tissue re-biopsy was suggested as appropriate procedure. Different from the study conducted by Kim et al., the population in this study were all Caucasian. I only had a minor concerns to authors.

Did any patient in this study received 3rd-generation EGFR-TKI such as osimertinib or CO-1686 later. If possible, can the authors showed the efficacy and outcome of the patients receiving 3rd-generation EGFR-TKI. 

Author Response

Dear Editor,

We are grateful for giving us the opportunity to revise our manuscript. We are grateful to all reviewers for their time and efforts  to improve the quality of our manuscript. In the revised version of our manuscript we have incorporated all suggested  comments from the reviewers. Please find below a point by point reply to the  comments: 

REVIEWER 2

This is a prospective study using plasma samples to detect the secondary EGFR-T790M mutation in advanced NSCLC patients with acquired resistance to first- and secondary-generations of EGFR-TKI. The EGFR-T790M positive rate in this study was 21.8% which was relative lower than that in previous studies.  A recent study  (Transl Lung Cancer Res. 2021 Feb;10(2):878-888.) showed similar results to this study and concluded that the sensitivity of plasma EGFR test for secondary T790M mutation in NSCLC with acquired resistance to EGFR-TKIs was low and tissue re-biopsy was suggested as appropriate procedure. Different from the study conducted by Kim et al., the population in this study were all Caucasian. I only had a minor concerns to authors.

Did any patient in this study received 3rd-generation EGFR-TKI such as osimertinib or CO-1686 later. If possible, can the authors showed the efficacy and outcome of the patients receiving 3rd-generation EGFR-TKI. 

Responses:

We thank the reviewer for these excellent comments. The indicated reference is now added in the discussion section (page 30), together with  a short statement comparing  the two populations (Caucasian vs Asian)

Regarding the use of Osimertinib on disease relapse, we would like to underline that analysis of subsequent treatments after disease progression was  not within the design or the scope of the current analysis. We are aware of the fact that osimertinib was administered mostly in patients with confirmed  T790M, but only the final clinical report of the study will provide adequate information on subsequent treatments and outcomes. We intend  to publish these results on subsequent treatment outcomes in a later publication

Reviewer 3 Report

Comments on MS "A real world observational study to access EGFR mutations upon progression on  or after 1st-line therapy with first or second generation EGFR TKIs in EGFR mutated positive NSCLC-LUNGFUL study" by Mountzios et al.

Following things need to be addressed in the MS:

1) Line 77 and line 576-author wrote activating EGFR mutations or sensitizing EGFR mutations of T790M-Should be explained more in detail in context to their role with first, second or third generation TKIs.

2) Line 520-author wrote plasma biopses can give false positives-very nice observation-should be discussed more how to rectify that technique in order to get the positive results as well as what could  have gone wrong when using that method?

The MS is well written with standard english. Comments above will make the MS better.

Author Response

Dear Editor,

We are grateful for giving us the opportunity to revise our manuscript. We are grateful to all reviewers for their time and efforts  to improve the quality of our manuscript. In the revised version of our manuscript we have incorporated all suggested  comments from the reviewers. Please find below a point by point reply to the  comments:

REVIEWER 3

Comments on MS "A real world observational study to access EGFR mutations upon progression on  or after 1st-line therapy with first or second generation EGFR TKIs in EGFR mutated positive NSCLC-LUNGFUL study" by Mountzios et al.

Following things need to be addressed in the MS:

1) Line 77 and line 576-author wrote activating EGFR mutations or sensitizing EGFR mutations of T790M-Should be explained more in detail in context to their role with first, second or third generation TKIs.

2) Line 520-author wrote plasma biopses can give false positives-very nice observation-should be discussed more how to rectify that technique in order to get the positive results as well as what could  have gone wrong when using that method?

The MS is well written with standard english. Comments above will make the MS better

Responses:

We would like to thank the reviewer for this very useful clarification. We have replaced the word “activating” with the word “sensitizing” in page 5 (lines 77 and 576). The reason is that activating mutations relate to ’gain-of-function’ mutations that change the gene product such that its effect gets stronger (enhanced activation) or even is superseded by a different and abnormal function. Sensitizing mutations are mutations that relate to the clinical response to a specific treatment such as osimertinib.

Based on findings from AURA studies (2& 3), specificity is 79% for the T790M, whereas specificity is not an issue for the activating EGFR mutations. The same finding appears in different  studies comparing T790M detection using  either cobas or different technologies. Genomic heterogeneity of T790M-mediated resistance may explain the reduced specificity observed with plasma-based detection of T790M mutations versus tissue. However, liquid biopsy is a valid option to consider when rebiopsy is not an option for the treating physician

 (Ref. 1. Takahama T, Azuma K, Shimokawa M, Takeda M, Ishii H, Kato T, Saito H, Daga H, Tsuboguchi Y, Okamoto I, Otsubo K, Akamatsu H, Teraoka S, Takahashi T, Ono A, Ohira T, Yokoyama T, Sakai K, Yamamoto N, Nishio K, Nakagawa K. Plasma screening for the T790M mutation of EGFR and phase 2 study of osimertinib efficacy in plasma T790M-positive non-small cell lung cancer: West Japan Oncology Group 8815L/LPS study. Cancer. 2020 Jan 1;126(9):1940-1948. doi: 10.1002/cncr.32749. Epub 2020 Feb 5. PMID: 32022929.

Ref. 2 Thress KS, Brant R, Carr TH, Dearden S, Jenkins S, Brown H, Hammett T, Cantarini M, Barrett JC. EGFR mutation detection in ctDNA from NSCLC patient plasma: A cross-platform comparison of leading technologies to support the clinical development of AZD9291. Lung Cancer. 2015 Dec;90(3):509-15. doi: 10.1016/j.lungcan.2015.10.004. Epub 2015 Oct 9. PMID: 26494259.

).  However, liquid biopsy is a valid option to consider when rebiopsy is not an option for the treating physician

All changes have  been highlighted in the revised manuscript for your  convenience

Reviewer 4 Report

The study is an observational study on EGFR T790M mutation detection in plasma from 96 NSCLC patients carried out by 23 hospital-based physicians in Greece. The EGFR testing was performed using the Cobas EGFR Mutation Test v2. EGFR T790M mutation was detected in 16% of the liquid biopsies. After re-biopsi was performed on 22% of the EGFR T790M negative patients the T790M positivity rate was 22%.

The topic is highly relevant and clinical interesting. However, as mentioned below do the manuscript lack some important information, so I don´t find it ready for publication in the current form.

Major comments:

Methods: There is little information about the pre-analytical steps for plasma sample collection e.g. which blood collecting tubes were used?, how much blood was collected?, how long was the time from plasma collection until plasma purification?, by which temperature were the plasma samples transported? how much plasma was used for the molecular testing? and how much ctDNA were obtained (min. and max ng) and do the samples contain genomic DNA? For the molecular testing, how was the analysis performed? How are the technical specifications for the Cobas test? (specially the sensitivity for the EGFR T790M mutation).

Methods: There are three whole pages containing the study design, study population etc. These parts need to be condensed and shortened; eventually can information be placed as supplementary information.

Table 5 and other places: In total 46 samples no mutations were detected, I find this number relatively high. What are the criteria for a conclusive analysis if the primary activating EGFR mutation is not present in the plasma sample? (is there enough cfDNA (and no genomic DNA) for detection of EGFR mutations?). This comment is connected to the one above on the preanalytical steps.

Minor comments:

Line 81+82+other places in the manuscript: Osimertinib, Erlotinib or Gefitinib -> osimertinib, erlotinib or gefitinib

Line 82 and 109: the ESMO abbreviation (line 109) is coming before the full name (line 89).

Line 84 and 574: “de novo” in italic font or not?

Table 4: Why is the ”R” symbol used?

Table 5: EGFRI-TKI -> EGFR-TKI

References: The reference list needs to be correct formatted for the journal e.g. Journal Articles:
1. Author 1, A.B.; Author 2, C.D. Title of the article. Abbreviated Journal Name YearVolume, page range.

Author Response

Dear Editor,

We are grateful for giving us the opportunity to revise our manuscript. We are grateful to all reviewers for their time and efforts  to improve the quality of our manuscript. In the revised version of our manuscript we have incorporated all suggested  comments from the reviewers. Please find below a point by point reply to the  comments:

REVIEWER 4

The study is an observational study on EGFR T790M mutation detection in plasma from 96 NSCLC patients carried out by 23 hospital-based physicians in Greece. The EGFR testing was performed using the Cobas EGFR Mutation Test v2. EGFR T790M mutation was detected in 16% of the liquid biopsies. After re-biopsi was performed on 22% of the EGFR T790M negative patients the T790M positivity rate was 22%.

The topic is highly relevant and clinical interesting. However, as mentioned below do the manuscript lack some important information, so I don´t find it ready for publication in the current form.

Major comments:

Methods: There is little information about the pre-analytical steps for plasma sample collection e.g. which blood collecting tubes were used?, how much blood was collected?, how long was the time from plasma collection until plasma purification?, by which temperature were the plasma samples transported? how much plasma was used for the molecular testing? and how much ctDNA were obtained (min. and max ng) and do the samples contain genomic DNA? For the molecular testing, how was the analysis performed? How are the technical specifications for the Cobas test? (specially the sensitivity for the EGFR T790M mutation).

Methods: There are three whole pages containing the study design, study population etc. These parts need to be condensed and shortened; eventually can information be placed as supplementary information.

Table 5 and other places: In total 46 samples no mutations were detected, I find this number relatively high. What are the criteria for a conclusive analysis if the primary activating EGFR mutation is not present in the plasma sample? (is there enough cfDNA (and no genomic DNA) for detection of EGFR mutations?). This comment is connected to the one above on the preanalytical steps.

Minor comments:

Line 81+82+other places in the manuscript: Osimertinib, Erlotinib or Gefitinib -> osimertinib, erlotinib or gefitinib

Line 82 and 109: the ESMO abbreviation (line 109) is coming before the full name (line 89).

Line 84 and 574: “de novo” in italic font or not?

Table 4: Why is the ”R” symbol used?

Table 5: EGFRI-TKI -> EGFR-TKI

References: The reference list needs to be correct formatted for the journal e.g. Journal Articles:
1. Author 1, A.B.; Author 2, C.D. Title of the article. Abbreviated Journal Name Year, Volume, page range.

Responses:

Methods:

  1. There is little information about the pre-analytical steps for plasma sample collection e.g. which blood collecting tubes were used?, how much blood was collected?, how long was the time from plasma collection until plasma purification?, by which temperature were the plasma samples transported? how much plasma was used for the molecular testing? and how much ctDNA were obtained (min. and max ng) and do the samples contain genomic DNA?

For blood collection, Streck Cell-Free DNA Blood CT blood collection tubes (cfDNA BCTs, (Streck, Nembraska, USA)) were used according to manufacturer’s instructions, in order to ship 10 ml of whole blood at ambient temperatures (15-25oC). Streck tubes were previously shown to retain the integrity of the cell-free DNA (cfDNA) population and stability of blood cells preventing dilution of circulating tumor DNA (ctDNA) with wild-type genomic DNA.

For the T790M testing, cobas® cfDNA Sample Preparation Kit and cobas® EGFR Mutation Test v2, were used for the isolation of circulating free DNA (cfDNA ) and the detection of EGFR mutations, respectively, as per manufacturer’s instructions. Briefly, 2 ml of plasma was used to extract cfDNA using cobas® cfDNA Sample Preparation Kit. The target DNA was then amplified and detected on the cobas z 480 analyzer (Roche Diagnostics, Pleasanton, USA) using the amplification and detection reagents provided in the cobas® EGFR Mutation Test v2 kit (Roche Diagnostics, Pleasanton, USA). Data analysis was automatically performed by EGFR Plasma Analysis Package Software version 1.0 (Roche Diagnostics, Pleasanton, USA)

Both kits are indicated by the FDA as Companion Diagnostics ( CoDx) for the TKI inhibitors erlotinib and osimertinib, in order to identify EGFR mutations (including  T790M) in the plasma of NSCLC patients

These comments are now included in the Materials and Methods Section

  1. For the molecular testing, how was the analysis performed? How are the technical specifications for the Cobas test? (specially the sensitivity for the EGFR T790M mutation).

For the molecular analysis, the cobas EGFR Mutation Test v2 (Roche Diagnostics) was used , a test bearing an FDA approval as a CoDx  for the TKI inhibitors erlotinib and osimertinib, in order to identify EGFR  mutations (incl. T790M)  in plasma from NSCLC patients. As per the analytical/non clinical evaluation of this test, a limit of detection of 100 copies/mL for T790M in plasma spiked with sheared cell line DNAs was reported.

As per its clinical evaluation, the  cobas® EGFR Mutation Test v2 for plasma, was used in the majority of clinical trials supporting Osimertinib’s use  in resistance T790M  setting, namely AURA extension and AURA2 phase II  and AURA III phase 3.. With cobas tissue test results as a reference, the plasma T790M positive percent agreement (PPA) was 61% and 51%  by cobas plasma, in AURA II and AURA III respectively.

  1. Table 5 and other places: In total 46 samples no mutations were detected, I find this number relatively high. What are the criteria for a conclusive analysis if the primary activating EGFR mutation is not present in the plasma sample? (is there enough cfDNA (and no genomic DNA) for detection of EGFR mutations?). This comment is connected to the one above on the preanalytical steps.

As shown in a recent exploratory ctDNA analysis, from patients enrolled in the  AURA 3 study, almost 25% of patients treated with Osimertinib were defined as non-shedders as they were not found positive for any of the 3 main EGFR mutations (T790M, del19, L858R) While there were no clear associations of the EGFRm tumor shedding status with sex, race, smoking status and  performance status according to Eastern Cooperative Oncology Group (ECOG) , an association  with the baseline tumor target lesion size has been reported 

In addition it has been previously reported that the detection of  EGFR mutations in both  plasma and tissue samples was associated with metastatic status . More specifically, the presence of extrathoracic (M1b) was associated with  the detection of EGFR mutations in plasma with statistical significance when compared to intrathoracic (M1a/M0) disease 

In the above context and taking into consideration that i. all molecular pathology laboratoties participating in the LUNGFUL trial used cobas EGFR Mutation Test v.2 for the detection of T790M in plasma (all these laboratories had previously participated in a quality control assurance study  for the detection of EGFR mutations in different plasma concentrations (Ntzifa at al) , and that ii. Streck Cell-Free DNA BCT blood collection tubes (cfDNA BCTs) were used to retain both the integrity of the cell-free DNA (cfDNA) population and the stability of blood cells preventing the dilution of circulating tumor DNA (ctDNA) with wild-type genomic DNA, one should exclude preanalytics and analytics steps for the inability to identify any mutation in the plasma of 46 patients.

Rather this should be attributed to either “non shedder” status or disease burden status as explained above.

There are three whole pages containing the study design, study population etc. These parts need to be condensed and shortened; DONE, Please see revised manuscript

Typing comments were addressed:

Line 81+82+other places in the manuscript: Osimertinib, Erlotinib or Gefitinib -> osimertinib, erlotinib or gefitinib  Done

Line 82 and 109: the ESMO abbreviation (line 109) is coming before the full name (line 89). Done

Line 84 and 574: “de novo” in italic font or not? Italic script was changed in normal script

Table 4: Why is the ”R” symbol used? ”R” was a typographical error that was deleted canceled (In our word format ‘R” appears as an arrow

Table 5: EGFRI-TKI -> EGFR-TKI  Done

References: The reference list needs to be correct formatted for the journal e.g. Journal Articles:
1. Author 1, A.B.; Author 2, C.D. Title of the article. Abbreviated Journal Name Year, Volume, page range." All references were introduced by EndNote software according to the journals’ requirements 

Round 2

Reviewer 1 Report

The Authors improved their manuscript, addressing the comments of the reviewers, but I keep thinking that the paper has a limed impact and it is inappropriate for this Journal. 

Reviewer 4 Report

The authors have answered my questions so I find that the manuscript is ready for publication. However, there are a number of spelling errors in the new text parts that need to be corrected.